# Improving the Efficiency of Precise Genome Editing with CRISPR/Cas9 to Generate Goats Overexpressing Human Butyrylcholinesterase

**DOI:** 10.3390/cells12141818

**Published:** 2023-07-10

**Authors:** Jia-Hao Wang, Su-Jun Wu, Yan Li, Yue Zhao, Zhi-Mei Liu, Shou-Long Deng, Zheng-Xing Lian

**Affiliations:** 1Beijing Key Laboratory for Animal Genetic Improvement, National Engineering Laboratory for Animal Breeding, Key Laboratory of Animal Genetics and Breeding of the Ministry of Agriculture, College of Animal Science and Technology, China Agricultural University, Beijing 100193, China; 2Department of Biomedical Engineering, College of Future Technology, Peking University, Beijing 100871, China; 3Laboratory Animal Center of the Academy of Military Medical Sciences, Beijing 100071, China; scauly@cau.edu.cn; 4NHC Key Laboratory of Human Disease Comparative Medicine, Institute of Laboratory Animal Sciences, Chinese Academy of Medical Sciences and Comparative Medicine Center, Peking Union Medical College, Beijing 100021, China

**Keywords:** CRISPR/Cas9, *FGF5*, *rhBChE*, homologous repair, SCNT

## Abstract

The CRISPR/Cas9 system is widely used for genome editing in livestock production, although off-target effects can occur. It is the main method to produce genome-edited goats by somatic cell nuclear transfer (SCNT) of CRISPR/Cas9-mediated genome-edited primary goat fetal fibroblast cells (GFFs). Improving the double-strand break (DSB) efficiency of Cas9 in primary cells would improve the homologous repair (HR) efficiency. The low efficiency of HR remains a major hurdle in CRISPR/Cas9-mediated precise genome editing, increasing the work required to screen the genome-edited primary cell clones. In this study, we modified several essential parameters that affect the efficiency of the CRISPR/Cas9-mediated knock-in GFF cloning system, including establishing a high-efficiency transfection system for primary cells via nucleofection and optimizing homology arm (HA) length during HR. Here, we specifically inserted a recombinant human butyrylcholinesterase gene (*rhBChE*) into the goat fibroblast growth factor (*FGF*)-5 locus through the CRISPR/Cas9 system, thereby achieving simultaneous *rhBChE* insertion and *FGF5* knock-out. First, this study introduced the Cas9, *FGF5* knock-out small guide RNA, and *rhBChE* knock-in donors into GFFs by electroporation and obtained positive cell clones without off-target effects. Then, we demonstrated the expression of rhBChE in GFF clones and verified its function. Finally, we obtained a CRISPR/Cas9-mediated rhBChE-overexpression goat.

## 1. Introduction

The type II bacterial CRISPR/Cas9 system, an RNA-guided nuclease system, is a robust tool for genome engineering in many cell lines and organisms [1,2,3]. The modules of this system are usually transfected into target cells via viruses or plasmid-dependent methods. Once this programmable nuclease is transferred into the cells, with the assistance of single guide RNA (sgRNA), it will generate site-specific DNA DSBs in the target locus to activate the DNA repair pathway [4]. Even if CRISPR/Cas9 efficiently causes the mutation of multiple cell lines and tissues in situ, the precise knock-in efficiency of the primary cells using homologous recombination (HR) is generally low [5,6]. Accordingly, it is crucial that we determine how to improve CRISPR-Cas9-mediated HR efficiency in primary cells.

Goat plays an essential role in scientific research of large economic animals, providing a series of high-quality products (such as meat, wool and milk). In addition, genome-edited goats have usually been selected as large livestock models, which are used for the genetics of breed improvement, studies of disease mechanisms, and production of agricultural products [7,8]. Using micro-injection of Cas9 protein and guide RNA into embryos, we antecedently obtained fibroblast growth factor 5 (*FGF5*) bi-allele knock-out sheep, with elevated sheep wool length and hair follicle density [9]. However, it remains an enormous challenge to the production of site-specific knock-in goats by microinjection. Somatic cell nuclear transfer (SCNT) of CRISPR/Cas9-mediated genome-edited primary fetal fibroblast cells is the main method to produce genome-edited goats [10,11,12]. For example, researchers had obtained overexpressed aralkylamine N-acetyltransferase (AANAT) in cloned goats, increasing the secreted level of endogenous melatonin as well as improving anti-inflammatory capacity [13]. Recently, some studies have determined that in vivo cleavage of longer genome-edited donors promotes CRISPR-mediated targeted integration in mice [14,15]. However, methods to produce cloned goats modified by DNA knock-in have not been optimized using new CRISPR/Cas9 approaches.

*FGF5* is a secreted signaling protein in time of the hair growth cycle in mammals, mutation of which can impact the anagen phase and result in new long-hair traits [16,17]. CRISPR/Cas9-mediated knock-out of the *FGF5* gene also promotes wool growth and increases the wool length and yield in goats, and a related study was reported to influence the coat color of goats [9,18]. Butyrylcholinesterase (BChE) is a non-specific esterase that is synthesized by the liver and released into the whole body through the blood [19]. It belongs to the serine hydrolase family, which shares a common tertiary structure with multiple spirals attached to the rope-folding core by multiple rear-helices [20,21,22,23]. BChE has a high clinical application value and can hydrolyze butyrylcholine and cocaine. It prevents nerve poisoning caused by the accumulation of neurotransmitters and treats organophosphorus poisoning [21]. The active site serine is covalently modified by a reaction with organophosphorus ester that simultaneously inactivates BChE and destroys the organophosphorus toxin [24].

In these experiments, in order to efficiently generate genome-edited cloned goats using CRISPR/Cas9 system, we established an efficient nucleofection program and optimized HA length in GFFs HR repair, enabling more efficient and precise gene editing in GFFs. Then, we attempted to combine CRISPR/Cas9 and SCNT to generate gene-edited goats that integrated *rhBChE* at the *FGF5* site and promoted rhBChE overexpression through *FGF5* knock-out. These gene-edited GFF clones also prevented nerve poisoning caused by the accumulation of 2,2-dichlorovinyl dimethyl phosphate (DDVP). In brief, we optimized the efficiency of homologous repair of primary GFFs and improved the productivity of gene-edited goats, demonstrating that the CRISPR/Cas9 system has the potential to become an important and applicable gene-editing tool in large animal breeding.

## 2. Materials and Methods

### 2.1. Animals

All animal experiments and treatments were approved and supervised by the Animal Welfare Committee of China Agricultural University (Permit Number: XK662).

### 2.2. Plasmid Construction

The *FGF5* HA-*rhBChE* donor was constructed as follows: firstly, total RNA was extracted from healthy donor peripheral blood mononuclear cells (PBMCs) using TRIzol Reagent (Invitrogen, Carlsbad, CA, USA). Then, RNA was directly used for cDNA synthesis with a first-strand cDNA synthesis kit (TaKaRa, Kusatsu, Shiga, Japan) based on the manual. The complete open reading frame (ORF) regions of the *BChE* gene were amplified using Q5 High-Fidelity DNA Polymerase (NEB). The ORF regions were assembled into the pUC57 donor and Sanger sequenced following the manufacturer’s instructions (Transgen, Beijing, China).

To construct a CRISPR vector for the FGF5 gene, pX330 and pX458 vectors containing Cas9 and U6-sgRNA co-expression backbones were purchased from Addgene. Then, we designed the goat FGF5 gene guide RNA (gRNA) sequences using the CRISPR Design website (http://tools.genome-engineering.org. Accessed on 5 August 2020). Two complementary gRNA oligos were cloned into the Cas9 backbone vector [25].

### 2.3. Cell Culture and Transfection

Goat fetal fibroblast cell lines (GFFs) were obtained from Lasoshan dairy goat (Capra hircus) fetuses at 6 to 7 weeks of pregnancy by the attaching tissue explant culture method. In short, the body of the goat fetus was washed three times with 1× Phosphate Buffer Saline (Gibco, Brooklyn, NY, USA) supplemented with penicillin–streptomycin (final concentration: 500 IU/mL, Gibco, Brooklyn, NY, USA). The body was broken down into small fragments (about 0.5 to 1 mm^3^ per fragment) and cultured in 60 mm Petri dishes with a complete medium containing DMEM/F12 medium (Gibco, Brooklyn, NY, USA) and 30% (*v*:*v*) fetal bovine serum (FBS, Hyclone, Logan, UT, USA) at 37.5 °C. Twelve hours later, we supplemented with 5 mL of DMEM/F12 medium containing 10% (*v*:*v*) FBS to the dish. When the GFFs were approaching 80–85% confluent, we would digest cells. Finally, primary GFFs were frozen and stored for future use.

For nucleofection, the Lonza Nucleofector 2b Device was used as outlined in the manufacturer’s instructions. Simply, 2.5 million GFFs were mixed with DNA plasmids or PCR products, which were resuspended in 100 µL buffer before nucleofection. In order to compare nucleofection efficiency, 10 mg pEGFP-N1 plasmid was used per experiment. For the knock-in experiments, we usually prepared about 10 µg pX330/PX458 vector and 5 µg amplified dsDNA repair template from the donor plasmid. After nucleofection, GFFs were seeded into six-well plates for further culture and analysis.

### 2.4. Sex Determination

Once primary GFFs formed cell lines, whether the sex of the GFFs was male or female needed to be determined via the sex-determining region of the Y-chromosome (*SRY*)-amplified in Appendix A. PCR-amplified products were used for electrophoresis analysis in an agarose gel.

### 2.5. Determination of Indel Frequency and PCR Genotyping

For *FGF5* gene knock-out experiments, primers ID-F/R were applied to PCR amplification around the targeted site’s region to determine the indel frequencies. After 48 h nucleofection, GFFs genomic DNA was extracted and the PCR amplification was performed within 10–50 ng DNA in 50 µL reactions. Then, PCR products with mutations were gel purified with a Gel Extraction Kit (Tiangen Biotech, Beijing, China) and subcloned into a pEASY-Blunt Zero Cloning Vector (TransGen Biotech, Beijing, China). Twenty colonies from each experiment were selected randomly for determining the indel frequency by Sanger sequencing.

### 2.6. Determination of Transfection/HDR Efficiency and Cell Sorting

For HDR efficiency experiments, *Cas9*-*FGF5* vector and repair template (HA-2A-mCherry) linearized with PCR amplified from the donor plasmid was prepared as follows: 10 µg PX330-*FGF5* and 5 µg HA-2A-mCherry template. After nucleofection, GFFs were seeded into 6-well plates for flow cytometry analysis. To detect the nucleofection efficiency and CRISPR/Cas9-mediated HDR efficiency, GFFs were digested and resuspended in 300 µL PBS with 3% BSA buffer. HDR efficiency data was obtained using a BD Fortessa flow cytometer [26].

For the establishment of positive GFF clones expressing rhBChE experiments, *Cas9-FGF5* vector and repair template (HA-CMV-*rhBChE*) linearized with PCR amplified from the donor plasmid were prepared. Forty-eight hours after nucleofection, GFP-positive single cells were sorted into 96 wells via BD FACS AriaIII. After 14 days of culturing, the cell clones were detected by *rhBChE* integration using PCR amplification.

### 2.7. Superovulation and Oocyte Collection

Firstly, it was determined that the female goat donors selected could weigh up to 40 ± 4.5 kg. Donor goats were injected with 300 mg progesterone by a controlled internal drug release (CIDR) device. To achieve superovulation, before the CIDR was removed from donor goats, they were injected 6 times with follicle-stimulating hormone (240 IU/dose) per 12 h intervals (Ning-bo Hormone Products Co; 110044648). In addition, a dose of 0.1 mg prostaglandin was injected when CIDR removal. And to induce ovulation, after CIDR withdrawal 38 h, 100 IU luteinizing hormone was injected. The oocytes were collected from the donor goat fallopian tubes according to previously established protocols. Finally, the number of all goat oocytes was recorded.

### 2.8. Somatic Cell Nuclear Transfer and Pregnancy Diagnosis

The qualified oocytes were transferred into TCM199 (Gibco, Brooklyn, NY, USA) with 2% (*v*:*v*) FBS. Mature oocytes were incubated in TCM199 medium (Gibco, Brooklyn, NY, USA) supplemented 5 μg/mL CB (Sigma-Aldrich, St. Louis, MO, USA; C6762) and 5 μg/mL Hochest 33,342 (Beyotime, Tongzhou, Beijing, China, C1022) about 5–10 min. Then, with the help of a micromanipulator, mature oocytes were enucleated and then rhBChE-positive GFFs were injected into the enucleated oocytes. After 90 min, these oocytes were performed electric fusion. Firstly, injected oocytes were transferred into a fusion solution for 3–5 min. Then, oocytes were placed into a fusion tank covered with the fusion solution.

The polar body of the oocyte was oriented parallel to the electrode and a DC pulse was applied 2 direct current pulses of 2.0 kV/cm for 25 μs using an ECM2001 Electrocell Manipulator (BTX Inc., San Diego, CA, USA). The fused embryos were transferred into TCM199 medium including 10 μg/mL of actinomycetes CHX (Sigma-Aldrich, St. Louis, MO, USA, C7698) and 5 μg/mL of CB for 4–5 h. Finally, the embryos were incubated to the developmental solution and transplanted the next day.

Sixty days after the embryo transfer, we would assess the pregnancy rate of the female goats via ultrasonography.

### 2.9. Transgenic Cloned Offspring

To determine the transgenic status of cloned goats, whole blood and ear tissue were collected for genomic DNA extraction. Clone goat has detected rhBChE integration using genomic PCR amplification (Appendix A). And insertion sequences were determined by Sanger sequencing.

### 2.10. RT-PCR Analysis

Total RNA was isolated with TRNzol Universal (Tiangen biotech, Haidian, Beijing, China; DP424) and rapidly reverse transcription by RT reagent Kit (TaKaRa; RR047A). The expression of the rhBChE cDNA gene was measured by qRT-PCR using the FastFire qRT-PCR PreMix (Tiangen biotech, Haidian, Beijing, China; FP207). Relative mRNA expression was calculated by the 2^−ΔΔCT^ method. The semiquantitative RT-PCR products were resolved in a 2.0% agarose gel for an electrophoresis analysis.

### 2.11. Western Blotting

For the GFFs, 1 × 10^6^ positive GFFs and WT GFFs were transferred into a 60 mm culture dish, then 10 µg/mL Brefeldin A (Beyotime Biotechnology) was added for 6 h. The cells were resuspended in PBS and centrifuged again. Then, the cells or tissues were lysed in RIPA lysis buffer (Solarbio, Tongzhou, Beijing, China) containing protease inhibitor cocktails (Beyotime Biotechnology). Briefly, the lysates were incubated on ice for 15 min, then centrifuged at 14,000 rpm for 10 min at 4 °C. Protein concentrations were determined by a BCA Protein Assay Kit (CWBIO) assay. Samples were separated by electrophoresis on SDS-PAGE gels and transferred onto polyvinylidene fluoride membranes (Solarbio). The samples were blocked for 120 min in Western Blocking Buffer (Solarbio). Then, the PVDF membranes were incubated overnight with mouse anti-GAPDH mAb (ZSGB-BIO) and BChE polyclonal antibody (Proteintech 23854-1-AP) (1:500). After being washed three times with TBST, PVDF membranes were incubated with horseradish-specific rabbit anti-goat IgG (H +L) (ZSGB-BIO) and horseradish-specific mouse anti-goat IgG (H +L) (ZSGB-BIO) for 60 min. Protein bands were visualized using the Bioanalytical Imaging System (Azure Biosystems, Dublin, CA, USA).

### 2.12. Fluorescence Microscopy

PX458 plasmid-transfected cells were seeded into 24-well plates, then stained with DAPI (Beyotime) for about 15 min and washed with PBS three times for 5 min. Images were taken via fluorescence microscope.

Next, 2.5 × 10^5^/well GFFs were cultured in 24-well plates for 12–16 h and then fixed with 4% paraformaldehyde and 0.1% Triton X-100. Cells were incubated with BChE polyclonal antibody (Proteintech) overnight at 4 °C, then incubated with Alexa Fluor 488-labeled goat anti-rabbit IgG (H + L). Images were taken by microscopy.

### 2.13. rhBChE Assay

rhBChE-overexpressing GFFs and WT GFFs were cultured in DMEM/F12 with 15% FBS until they were up to 70–90% confluent, then the medium was changed with Opti-MEMTM (Gibco, Brooklyn, NY, USA) for 1 day. The rhBChE in the supernatant was detected using the Butyrylcholinesterase assay kit (Nanjing Jiancheng Bioengineering Institute). After the addition of organophosphorus pesticides, GFFs were seeded in 96-well plates to evaluate the MID (median lethal dose) of Dichlorvos in acetone (GBW 081320). Then, rhBChE and WT GFFs were seeded separately in 96-well plates and cultured in DMEM/F12 with 15% FBS adding the MID of Dichlorvos for 24 h. Then, cell viability was detected by an MTT Cell Proliferation and Cytotoxicity Assay Kit (Solarbio, Tongzhou, Beijing, China) in accordance with the manufacturer’s instructions.

### 2.14. Statistical Analysis

Data were analyzed by Student’s *t*-test. Data are expressed as mean ± standard deviation (SD). The differences were considered statistically significant at *, *p* < 0.05; **, 0.001 < *p* < 0.01; and ***, *p* < 0.001.

## 3. Results

### 3.1. Establishing a High-Efficiency Nucleofection Method and Detecting the Indel Efficiencies of FGF5 sgRNAs in GFFs

Three primary GFFs from dairy goat fetuses were isolated by the attaching tissue explant culture method (Appendix A). Polymerase chain reaction (PCR) of the *SRY* gene revealed that GFF #1 was female, and GFFs #2 and #3 were male (Appendix A, Appendix A). In subsequent experiments, we used the male GFF #2 for all studies.

The low transfection efficiency of GFFs was the main factor limiting genome-editing research in clone goats. First, we selected a guide RNA in the third exon of the goat *FGF5* gene (Figure 1a) and introduced the sgRNA sequence into the Cas9-GFP plasmid, which was used to screen gene-edited cells in this study (Figure 1b). To explore how to achieve high transfection efficiency in fibroblast cells, we contrasted two different transfection methods, lipofection as well as nucleofection via an EGFP-N1 plasmid. After 48 h transfection with the EGFP-N1, the transfection efficiency of lipofection of the GFFs approached 4.73% ± 0.5% and the nucleofection efficiency (program A033) was 52.06% ± 3.2%, much higher than the liposome transfection (Figure 1c,d). Next, because the nucleofection efficiency was significantly higher than other methods, we tried to optimize appropriate procedures to achieve higher indel efficiency and cell viability. We selected several nucleofection programs that might be suitable for GFFs. After GFFs were nucleofected with the Cas9-*FGF5* guide RNA vector, genomic DNA of different programs were, respectively, extracted and the PCR amplifications were performed with primers ID-F/R (Appendix A). Sanger sequencing assays showed the atlas of *FGF5* target sites (Figure 1e, left). And TA cloning assays showed that the indel rates were between 5% to 35% (Figure 1e, right). It is suggested that the guide RNA we designed can produce higher indel rates. Of these, A033 had the highest transfection efficiency (48%) and indel frequency (35%) after GFF transfection (Figure 1e and Appendix A). Finally, we selected nucleofection as the suitable transfection method for GFFs, and we used A033 as the GFF nucleofection program in all subsequent experiments.

### 3.2. Optimizing HA Length during HR in GFFs

To enhance the HR efficiency at the *FGF5* locus in GFFs, we subsequently optimized the HA length of the repair template donor. We first established a fluorescent report knock-in system at the *FGF5* locus in GFFs (Figure 2a), after 2 days of co-transfection (pX330-*FGF5* vector and *FGF5* HA-T2A-mCherry repair donor). Then, we determined the percentage of mCherry fluorescence in *FGF5* target-specific integration of HA lengths from 100–1000 bp via flow cytometry. When the HA length was at 800 bp and 1000 bp, the homologous repair efficiency reached 2.58% and 2.96%, respectively. This is an extremely significant difference to the repair efficiency of 100 bp HA templates and a significant difference to other HA templates (Figure 2b,c, Appendix A). There was no significant difference in mean fluorescence intensity (MFI; Figure 2d) because the promoter of the *FGF5* gene expressing the mCherry fluorescence protein is endogenous. Obviously, the HA length is an essential factor affecting HR efficiency. When the length of the HA was 800–1000 bp, the efficiency of homologous repair was higher in GFFs.

### 3.3. Establishment of Positive Goat Fibroblast Cell Clones Expressing rhBChE

The rhBChE cDNA was amplified from healthy donor peripheral blood mononuclear cells (PBMCs). The rhBChE expression donor (containing the CMV promoter, SV40 polyA, and ~1000 bp HA) and the *FGF5* sgRNA-Cas9 T2A GFP expression vector were nucleofected into GFFs, which might complete *FGF5* site-specific insertion of *rhBChE* (Figure 3a). Single cells expressing GFP were inoculated into 96-well plates for culture using flow cytometry sorting. After approximately 2 weeks, every single clone was identified. Single-cell clones were determined by PCR-amplified and Sanger sequencing with site-specific insertion of *rhBChE* at the *FGF5* locus (Figure 3a, Appendix A). To detect the insertion of genes, two pairs of primers were designed at both ends of the fragment. The 3′ upstream primer RI-F was located on the SV40 polyA sequence of the inserted fragment. The downstream primer RI-R was located outside the 3′ homologous arm sequence and inside *FGF5* (amplification product, 1655 bp; Figure 3d, Appendix A). Similarly, the 5′ primers LI-F and LI-R were located inside the *FGF5* gene and the CMV sequence (amplification product, 1482 bp). Insertion sequence locations were determined by Sanger sequencing (Figure 3b,c). The genomic integration of *rhBChE* in single-cell clones was identified by cross-homologous arm PCR using ID-F/R as upstream and downstream primers (amplification product, 3004 bp; Figure 3e, Appendix A).

In brief, as displayed in Table 1, PCR amplified showed that 15 positive cell clones had simultaneous 5′ and 3′ integration at the *FGF5* monoallelic level, indicating an integration efficiency of 9.15% (where specific integration efficiency = no. of cell clones with specific integration/no. of total cell clones × 100%). Based on the morphology and growth of the cell clones, three positive cell clones were selected for subsequent experiments. First, we identified the *rhBChE* expression in positive cell clones #10, #29, and #43 and negative wild-type cell clones. The results showed that *rhBChE* mRNA was expressed in the three positive cell clones, and the level of *rhBChE* in cell clones #10 and #29 was significantly different from wild-type cells (Figure 3f,g). Subsequently, the protein expression of rhBChE in positive cell clone #10 was detected by immunofluorescence (Figure 3h). Finally, we selected clone #10 as the donor cell for nuclear transfer.

### 3.4. Nuclear Transfer for Generating rhBChE Knock-in Goats

We collected 118 MII oocytes from 10 female goats using multiple superovulation, of which 93.22% (110/118) oocytes qualified. The donor cell SCNT enucleation rate was 94.55% (104/110) and the fusion rate was 85.58% (89/104). The fused embryos were transferred into eight recipient female goats, of which the pregnancy rate was 12.5% (Table 2). Ultimately, one recipient delivered (Figure 4a). Ear tissues from the *rhBChE* overexpression and WT goats were collected. In order to detect target-specific integration of the exogenous rhBChE gene, we extracted tissue DNA for PCR amplification (Figure 3a). The 1482 bp fragments were amplified using primers LI-F/R, and Sanger sequencing showed that the *rhBChE* donor was inserted into exon 3 of the *FGF5* gene (Figure 4b,c, Appendix A). Full-length *rhBChE* in the goats was identified by cross-homologous arm PCR using ID-F and ID-R, in which the amplification product was 3004 bp (Figure 4d). In particular, Sanger sequencing showed that 5′ upstream of the *rhBChE* donor was inserted into exon 3 of the *FGF5* gene (Figure 4e). However, the sequencing results showed that the cloned goat had simultaneous *FGF5* knock-out and *rhBChE* integration at the monoallelic level (Figure 4e). We also detected an indel of another *FGF5* allele chain, with the insertion of an A base into the *FGF5* target (Figure 4f). Translation of both the double strands was broken at the *FGF5* locus (Figure 4g).

To check whether there were off-target effects in the genome of the cloned goat, we determined other potential off-target sgRNA sequences of Cas9 [27]. These off-target sequences both ended before NGG bases and aligned most effectively with the guide RNA. Subsequently, we selected the top ten off-target sequences as shown in Figure 4h, and no off-target effects were found.

### 3.5. rhBChE Protein Expression and Functional Assays

To explore whether the *rhBChE* we integrated into the cloned goat was successfully expressed, we determined the expression of rhBChE using Western blot. We collected peripheral blood and skin tissue of the clone goat one month after the birth. The clone goat sample was obtained by extracting the total protein of the skin tissue. The expression level of rhBChE of a cloned goat was obviously increased compared with the WT goat (Figure 5a). It was not remarkably different in GAPDH levels between the two animals. However, the only resulting transgenic goat died unexpectedly a year after birth, so we did not have time to identify other functions and traits.

To identify the expression of extracellular rhBChE, we examined the acetylcholinesterase activity via Ellman colorimetry assay, of which the enzyme source was the cell clone supernatant. The results showed that the positive cell clones #10, #29, and #43 were determined to have high acetylcholinesterase activity, and the levels of rhBChE in all three cell clones were significantly higher than wild-type cells (*p* < 0.001) (Figure 5b). To test whether the rhBChE-positive cells had the ability to resist organophosphorus pesticide injury, DDVP (an organophosphorus pesticide) was added to the cell culture medium to cause damage to cells, and the cell proliferation and cytotoxicity were detected using the MTT kit. First, DDVP was added to WT cells to evaluate the MID of DDVP in a dose-dependent manner. The results showed that the IC50 of DDVP in the WT clone was 145.5 µM (Figure 5c). Then, 145.5 µM DDVP was added into positive cell clone #10 and the WT clone, which found that positive cell clone #10 had better cell proliferation ability. The difference in the WT cells was significant (*p* < 0.05). The results showed that the rhBChE-positive GFF cell clone had better resistance to DDVP damage (Figure 5d).

## 4. Discussion

With the emergence of the CRISPR/Cas9 system, the progress of genome editing of large animal models has accelerated tremendously. This technology only needs an artificial sgRNA (including a PAM sequence) to guide the Cas9 nucleases to break any target sequence in the large animal genome. Then, the Cas9-sgRNA complex can produce target-specific DSBs, which can activate the DNA self-repair mechanism such as NHEJ. Subsequently, NHEJ leads to targeted mutagenesis at targeted sites [15,28]. In our study, we used a previously reported sgRNA sequence that can produce an efficiently edited *FGF5* gene in GFFs, and then NHEJ generated a frameshift of gene expression. The CRISPR/Cas9 system can also edit multiple targets via the pronuclei microinjection of several guide RNAs, as well as Cas9 protein into zygotes to establish animal models of multigenic diseases, such as complex Alzheimer’s disease (AD) [8,29]. Therefore, large animal models can better simulate the genetic mutations and symptoms of patients and provide an effective treatment basis for genetic diseases. Furthermore, the CRISPR/Cas9 technology is able to improve livestock performance, of which one of the most representative experiments involves the knock-out of myostatin (MSTN). As a result of hyperplasia and hypertrophy of muscle fibers, the knock-out cows resulted in the double muscling phenotype [30,31]. These genetically modified animals have superior muscle mass compared with their non-modified counterparts [31].

To our knowledge, there are few studies where site-specific integration has been applicated using CRISPR/Cas9 system via the NHEJ and HR pathways in goats, particularly when human proteins were integrated into the goat genome and expressed successfully. In general, our research demonstrates that both NHEJ- and HDR-mediated genome editing can be performed successfully in GFFs. In addition, we constructed a non-resistant and unmarked knock-in overexpressing a rhBChE-HA donor and obtained 17 positive GFF clones by PCR and Sanger sequence screening other than using fluorescence or drug. Due to no marker genes being expressed in the donor cells, there is no potential biosafety risk for the generation of transgenic animals. However, the only cloned goat that resulted died unexpectedly a year after birth, so we did not have time to identify other functions and traits of the cloned goat.

SCNT is an effective means to produce cloned animals, and primary fibroblast cells are the major donor cells for SCNT. However, progress in the site-directed modulation of large animal gene loci is restricted by the low efficiency of genome editing, imposed by the low efficiency of HR. In addition, the limited proliferative capacity of primary somatic cells is also an important influencing factor [32]. Several studies have reported that the use of single-stranded oligodeoxynucleotides (ssODN) as the repair template exhibits higher HDR efficiency than dsDNA [15,33]. However, such a strategy is limited to the introduction of short edits (<50 nt). In contrast, if we intend to site-integrate a whole protein DNA sequence in a large animal genome, dsDNA is suitable for introducing large sequence changes and insertions, which broadens the range of applications. Although the length of our insert fragments was more than 3000 bp, efficient integration can be achieved through long dsDNA homologous arms in the goat cell clones.

While the CRISPR/Cas9 system is rapidly advancing research in large animal models, the risk of off-target still remains a controversial issue in this technology [34,35]. Up to now, there are no definitive rules that can accurately predict Cas9 specificity, but some research has been proposed to avoid off-target cleavage events, such as several specificity Cas9 mutants (Cpf1, Cas13d, SaCas9 and XCas9) [35,36,37]. In our study, we also evaluated off-target effects via the Cas-OFFinder software to predict the possible off-target sites and detect the top ten most likely sites. Although no mutations were found, high-throughput genome-wide assays are needed to confirm whether any have occurred off-target.

## 5. Conclusions

In summary, we proved that CRISPR/Cas9 can be applicated for genome editing at a selected target site in the primary GFFs. In addition, using a non-resistant and unmarked knock-in rhBChE-overexpressing donor, we obtained positive clonal cell clones by PCR-amplified screening. Finally, we also obtained a CRISPR/Cas9-mediated rhBChE-overexpressing goat. Our research further demonstrates that CRISPR/Cas9 technology has broad applicability in large animal cloning.

## Figures and Tables

**Figure 1 cells-12-01818-f001:**
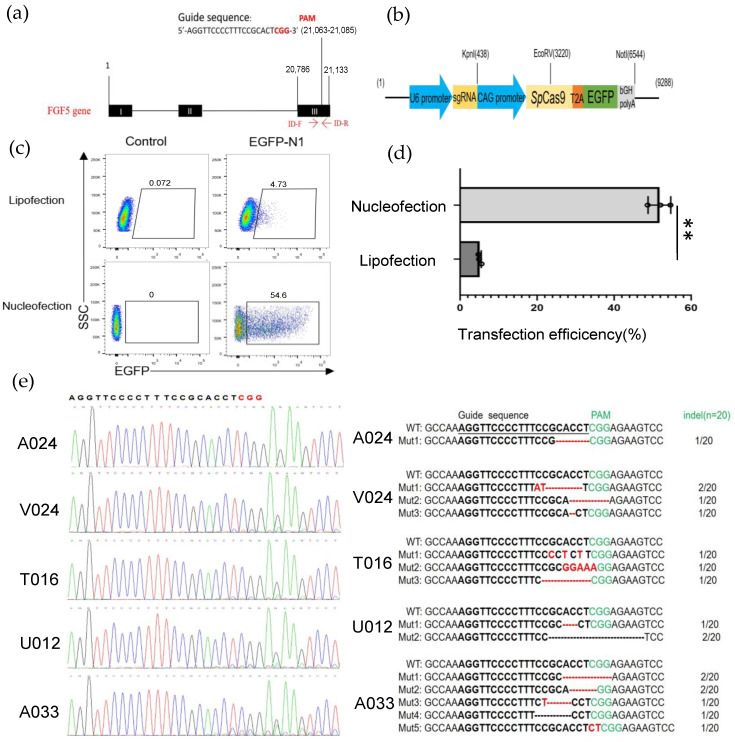
Establishing an efficient nucleofection method in GFFs. (**a**) Schematic of genome editing at the *FGF5* locus; (**b**) the construction of sgRNA-Cas9 expression vector; (**c**) detection of the transfected efficiencies of lipofection and nucleofection by flow cytometry after 2 days transfection; (**d**) effect of the efficiencies of lipofection and nucleofection after 48 h transfection by a histogram, (n = 3 biological replicates). Significance was calculated using *t*-test: ** *p* < 0.01; (**e**) TA cloning and Sanger sequencing analysis of the mutation types and indel frequencies determined in GFFs transfected with Cas9 plasmid. Deleted nucleotides are shown by a dashed line (-), and fresh-added nucleotides are emphasized by red letters. Indel rates are behind the bases.

**Figure 2 cells-12-01818-f002:**
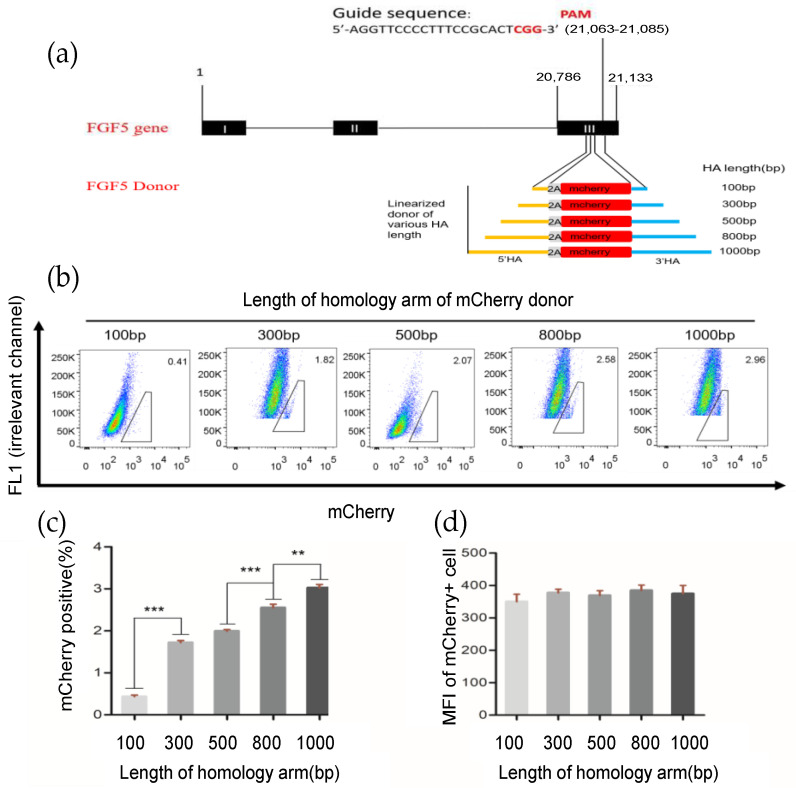
Effects of different length homologous arms on homologous repair efficiency. (**a**) Schematic outline of mCherry HDR donor with *FGF5* locus homologous arms ranging from 100–1000 bp in length; (**b**) determination of the HDR efficiency by FACS. The percentages of mCherry^+^ cells represent the HDR efficiencies; (**c**,**d**) HDR efficiency and mean fluorescence intensity with different HA lengths. n = 3 biological replicates; ** *p* < 0.01; *** *p* < 0.001.

**Figure 3 cells-12-01818-f003:**
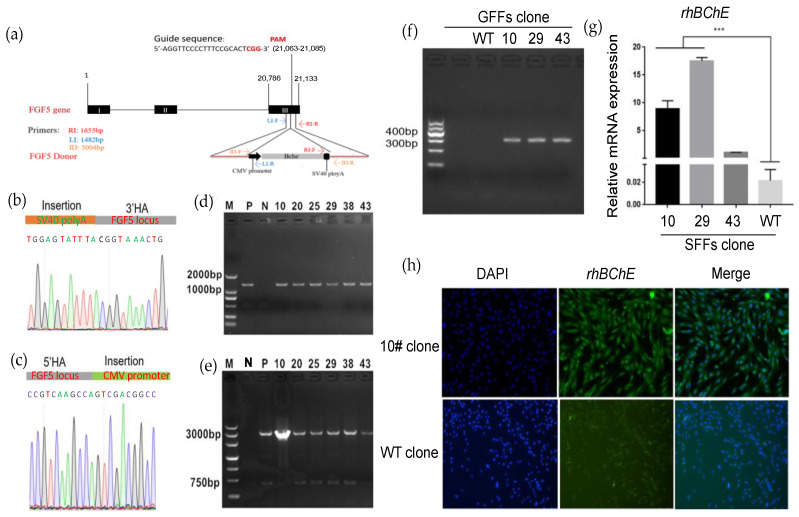
Construction of goat GFF single-cell clones expressing rhBChE at the *FGF5* locus via HR. (**a**) Experimental outline and schematic of the HR process. Both *FGF5*-sgRNA/Cas9 plasmid and HR arm-CMV-*rhBChE* template were co-transfected into GFFs; (**b**) Sanger sequencing of the 3′ junction regions (CMV-*rhBChE*) upon correct targeting at the *FGF5* locus. The orange area shows the HR template before the *FGF5* DSB site; (**c**) Sanger sequencing of the 5′ junction regions (CMV-*rhBChE*) upon correct targeting at the *FGF5* locus. The green area shows the CMV promoter after the *FGF5* DSB site; (**d**) the 3′ junction PCR analyses confirming the site-specific targeting in the GFF clones expressing rhBChE by Primer RI-F/R. P, positive mix cell clone; N, negative wild-type clone; others, positive GFF gene-editing clones (5′ junction PCR analyses are shown in Appendix A); (**e**) identification of full-length *rhBChE* donor in the GFF clones expressing rhBChE via ID-F/R; (**f**) detection of expression of rhBChE by cDNA-PCR; (**g**) the RNA expression of rhBChE in GFF gene-edited single-cell clones; (**h**) single-cell clones were identified using immunofluorescence assays with rhBChE antibodies and then were observed by fluorescence microscopy. *** *p* < 0.001.

**Figure 4 cells-12-01818-f004:**
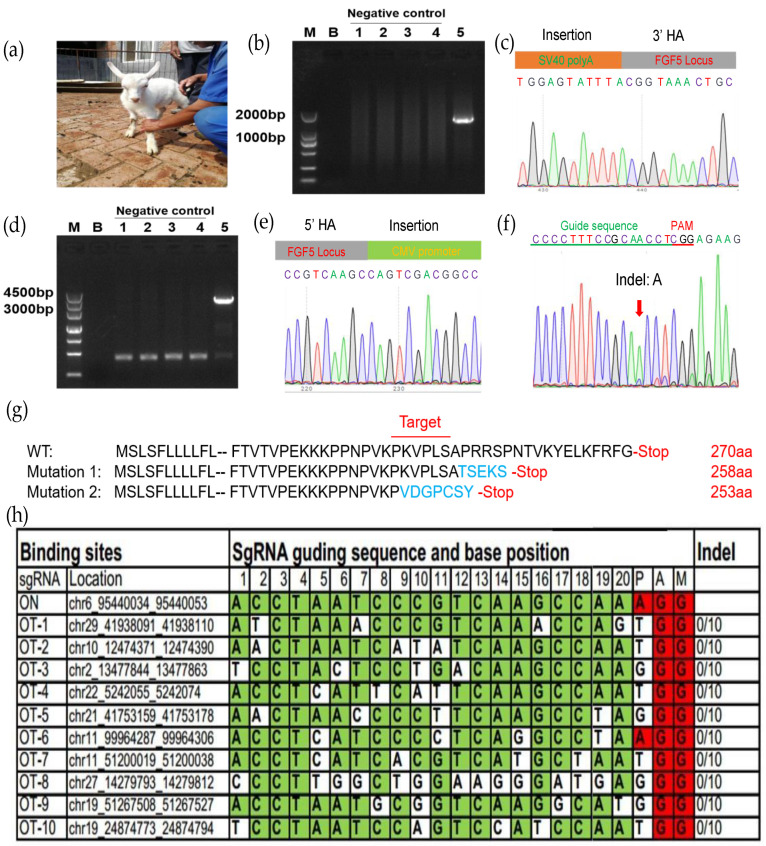
Assessment of the transgenic goat. (**a**) Picture of one-month-old goat carrying *rhBChE* integration; (**b**) the 3′ junction PCR results from the transgenic goat; 1–4, control negative one-month-old WT goat; 5, positive clone goat. The PCR templates were genomic DNA that came from the goat ear tissue; (**c**) Sanger sequencing of the 3′ junction regions upon correct targeting at the *FGF5* locus; (**d**) full-length PCR results from 5′ HA to 3′ HA via primer ID-F/R from transgenic goat; 1–4, control negative one-month-old WT goat; 5, positive clone goat; (**e**) Sanger sequencing of the 5′ junction regions upon correct targeting at the *FGF5* locus; (**f**) Sanger sequencing of another NHEJ DNA chain region at the *FGF5* locus; (**g**) schematic diagram of amino acid changes at the target site at modified FGF5 locus in cloned goat. Amino acid variations of the sgRNA-targeting site are shown in blue; (**h**) top ten off-target sites of sgRNA *FGF5*. OT, off-target. Green colors represent the same base at the target site; red colors represent the same base at the PAM site.

**Figure 5 cells-12-01818-f005:**
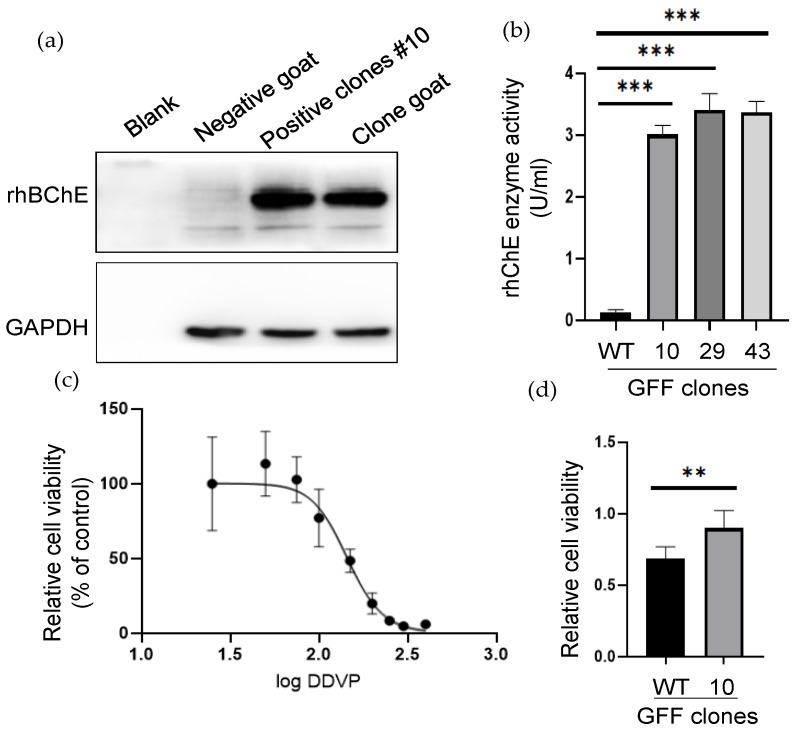
The rhBChE expression and functional analysis of positive GFF clones. (**a**) Western blot analysis of the rhBChE expression of the #10 monoclonal cell and cloned goat. Negative control, one-month-old WT goat. The clone and WT goat samples were obtained from skin tissue; (**b**) analysis of supernatant protein activity. The activity of supernatant proteins after ultrafiltration of the different cells was determined by the Ellman assay. Error bars, SD from three technical replicates. Significance was calculated using Student’s unpaired *t*-test, *** *p* < 0.001; (**c**) determination of anti-DDVP ability of WT clone. Next, 145.5 µM was the median lethal dose (MID) of the WT clone; (**d**) Analysis of anti-DDVP ability of WT and positive clone after 145.5 µM DDVP stimulation. Error bars, SD from three technical replicates. Significance was calculated using Student’s unpaired *t*-test, ** *p* < 0.05.

**Table 1 cells-12-01818-t001:** Summary of the generation of the goat fibroblast cell line expressing rhBChE.

Batch Number	No. of Single Colonies	No. of 5’ Junction PCR Colonies (%)	No. of 3’ Junction PCR Colonies (%)	Senesced ^a^	Suitable Colonies for SCNT
1	64	6 (9.68%)	6 (9.68%)	3	3
2	58	5 (8.62%)	5 (8.62%)	3	2
3	42	6 (14.2%)	5 (11.90%)	4	1

a Colonies were scored as senesced when cell number were not observed to increase seven days after seeding.

**Table 2 cells-12-01818-t002:** Summary of the nuclear transfer results from GFFs overexpressing rhBChE.

No. of Donor Goats	No. of Collected Oocytes	No. of Usable Oocytes	Oocytes Enucleated Rate (%)	Fusion Rate (%)	No. of Recipient	Pregnancy Rate (%)	Lambing Rate (%)
10	118	110	94.55 (104/110)	85.58 (89/104)	8	12.5 (1/8)	100 (1/1)

## Data Availability

All data are available in the main text and the Appendix A.

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
