# Peer review of "Improving the Efficiency of Precise Genome Editing with CRISPR/Cas9 to Generate Goats Overexpressing Human Butyrylcholinesterase"

_cells, 2023, doi:10.3390/cells12141818_

Round 1

Reviewer 1 Report

The manuscript by Wang et al., describes the knockin of Butyrylcholinesterase (BChE) into the FGF5 gene of goat in conjunction with FGF5 knockout. This was achieved by CRISPR/Cas9 engineering in fetal fibroblast cells and subsequent somatic cell nuclear transfer.

This study examines the impact of two transfection approaches, and evaluates appropriate homology arm length (100 – 1000 bp). However, within the field of genome engineering/integration, the optimisation of parameters have been extensively studied.

Therefore the novelty in the article principally lies in the use of species to engineer and the amalgamation of genome editing and animal cloning technologies. A functional evaluation of BChE-positive cells to resist organophosphorus pesticide injury was made. It was misfortunate that the traits of the single live birth transgenic goat could not be examined in detail — the animal died a year after birth, precluding downstream studies.

Comments/suggestions

-As mentioned above, the only negative I would have with this manuscript is with novelty. CRISPR/Cas9 parameter optimisation is a very heavily published area and what this study brings is mostly an approach that works in SCNT in goat.

-The manuscript is well written (with some minor changes suggested below). It was a pleasure to read.

-The methods are clear and of sufficient detail. However, in the section that describes HDR efficiency its stated that ‘10 mg PX330- 132 FGF5 and 5 mg HA-2A-mCherry template’ was used. Presumably, this should be micrograms and not milligrams.

-In the Results section 3.1 these is a spelling mistake where is states, “Thus, we chose the ucleofec”.

-In the discussion, it is stated, “In addition, we constructed a non-resistant and unmarked knock-in overexpressing an rhBChE-HA donor (Figure 3a) and obtained many positive clonal cell clones by blind screening. Thus, when no marker gene remains in the donor cell, there is no potential risk for the generation of transgenic animals.” This description is not clear and needs to be modified.

The English is fine with a few minor corrections as highlighted in the response to the authors

Author Response

Comments/suggestions

-As mentioned above, the only negative I would have with this manuscript is with novelty. CRISPR/Cas9 parameter optimisation is a very heavily published area and what this study brings is mostly an approach that works in SCNT in goat.

Response: Thank you for your questions. The main novelty of this paper is we established an efficient nucleofection system and optimized HA length during HR optimized of GFFs, enabling more efficient precise gene editing in GFFs. Then, we attempted to combine CRISPR/Cas9 and SCNT to generate gene-edited goats that integrated rhBChE at the FGF5 site and promoted rhBChE overexpression through FGF5 knockout. Accoroding to previous reports, the extremely low efficiency of homologous recombination (HR) and the time-consuming breeding programs were required to obtain genetically modified animals. The progress of genetic modification in large animals is also slow going since zygote injection, combined with inefficient HR, limits the possibility to achieve precise and tailored editing (Zhao, et al., 2019, Natl Sci Rev, 402-420). We modified several key parameters which improved the efficiency of the CRISPR/Cas9-mediated knock-in GFF cloning system and our results showed that we made a significant improvement.

-The manuscript is well written (with some minor changes suggested below). It was a pleasure to read.

Response: Thank you for your positive comments.

-The methods are clear and of sufficient detail. However, in the section that describes HDR efficiency its stated that ‘10 mg PX330- 132 FGF5 and 5 mg HA-2A-mCherry template’ was used. Presumably, this should be micrograms and not milligrams.

Response: Thank you for your careful reading. Accordingly, we have revised the whole section and changed “10 mg PX330- 132 FGF5 and 5 mg HA-2A-mCherry template” into “10 μg PX330- 132 FGF5 and 5 μg HA-2A-mCherry template” in the revised manuscript (line 131-132).

-In the Results section 3.1 these is a spelling mistake where is states, “Thus, we chose the ucleofec”.

Response: Thank you for highlighting this mistake and we apologize for the careless writing. We have corrected the word “ucleofection” to “nucleofection” in the revised manuscript (line 268).

-In the discussion, it is stated, “In addition, we constructed a non-resistant and unmarked knock-in overexpressing an rhBChE-HA donor (Figure 3a) and obtained many positive clonal cell clones by blind screening. Thus, when no marker gene remains in the donor cell, there is no potential risk for the generation of transgenic animals.” This description is not clear and needs to be modified.

Response: Following the reviewer’s suggestion, we have replaced the mentioned part in discussion into “In addition, we constructed a non-resistant and unmarked knock-in overexpressing an rhBChE-HA donor and obtained 17 positive GFF clones by PCR and sanger sequence screening other than using fluorescence or drug. Due to no marker genes in the donor cells, there is no potential biosafety risk for the generation of transgenic animals.” (448-452).

Reviewer 2 Report

Dear Editor,

The article by Wang et al. is a beautiful example of CRISPR-Cas9 gene editing in live-stock. The authors carefully performed all the crucial steps to reach their goal of a goat overexpressing rhBChE. Included are transfection methods, plasmids used, targeting analysis, optimal homology arm length, screening and immunofluorescence. It will be a guideline for scientists who want to perform a similar approach. 

Although not crucial, it would be interesting if the authors have some information as to whether the animal died a year after birth (post-mortem-examination). Is this related to the overexpression of rhBChE, related to FGF5-knock out, related to the method (off-target) or bad-luck. Is survival normal higher (hard to assess in case n=1).  This could be essential information for researchers wanted to copy this approach.

Author Response

Although not crucial, it would be interesting if the authors have some information as to whether the animal died a year after birth (post-mortem-examination). Is this related to the overexpression of rhBChE, related to FGF5-knock out, related to the method (off-target) or bad-luck. Is survival normal higher (hard to assess in case n=1).  This could be essential information for researchers wanted to copy this approach.

Response: Thank you for careful revision and posing this important question. The rhBChE-overexpression goat obtained in this study died a year after birth, then a simple autopsy was performed and no significant histological lesions were found. Other study has reported that the rhBChE-overexpression goats produced milk with high concentration of BChE at 10 months of pregnance, without mentioned the death of goats (Huang, et al., 2007, P Natl Acad Sci USA, 13603-13608), so we speculated that rhBChE overexpression would not cause death of the goats.

In addition, FGF5 single gene edited goats were prepared by our laboratory before which maintained good growth condition and good reproductive performance as normal goats. Therefore, we believed that the death of gene edited goats in our experiment was not related with FGF5 gene editing.

Moreover, following the suggestion from the reviewer, we have checked the off-target effects in the genome of the goat overexpressing rhBChE (line 368-370), and no off-target signal was found among the top ten off-target sites. Combined with the current situation, we speculated that the death of the sheep may be related to late feeding management. However, as stated by the reviewer, it is hard to assess survival at n = 1.

Reviewer 3 Report

In the paper “Improving the efficiency of precise genome editing with CRISPR/Cas9 to generate goats overexpressing human butyrylcholinesterase”, the authors claim to show an improvement in HR in goat fibroblasts and consequently, production of an animal bearing the desired modification after SCNT. In my opinion, the data produced is not new, since transgenic goats expressing the same transgene were already produced, and different methods to improve HR were also already tested. For example, the use of DNA repair molecules or their stimulatory compounds were tested to improve HDR efficiency in many cell types and animal models, including goats. Goats that expressed rh butyrylcholinesterase were also produced, and the production of the recombinant protein was under a milk promoter, what would facilitate the recovery of the target protein. 

Here are some concerns that should be addressed by the authors. 

1)    Please elucidate how would the protein be recovered from the transgenic animals?

2)    The main objective of this paper is not clear. Is it to improve HR efficiency? Produce a transgenic animal? Proof of concept regarding the use of CRISPR/Cas system to generate transgenic large animals?

3)    Please clarify the specie from which BChE was extracted from. I believe it was from a human. Was there an ethic committee involved? If so, please indicate. 

4)    Be careful when describing units. From lines 103-114, all the measurements were in the milli scale (ml and mg). Is that correct? If so, please elaborate how feasible would be to produce plasmids and DNA template constructs in the “mg” scale. Same thing for line 126, 132-133.

5)    During oocyte manipulation (cloning procedure), no protein source was added into the TCM199?

6)    Please clarify the methodology used for oocyte activation after SCNT. Has the stated protocol (cycloheximide for 5 min followed by cytochalasin B for 5hrs) been used before? If so, please add a reference. 

7)    Why was melatonin used in different concentrations or sometimes not even used? Do the authors have any results showing the effects of melatonin on embryo survival/transfer efficiency?

8)    Did the genetic manipulation of the cell affect embryo development rates? No results were shown regarding in vitro embryo development.

9)    Did the authors had WT SCNT animals born? Were CT embryos transferred to recipient females?

10) The introduction is very broad and with very overstated phrases, such as lines 73-76. The same happened during the discussion and conclusion. The discussion session does not bring any discussion into the presented results. 

11) Line 269: Figure 1f doesn’t exist. 

12) Figure 2 c and d: odd scale

13) Please clarify which of the cell clones colony had biallelic modification. Was this the colony used for SCNT?

14) Table 1: Please be careful when describing number of 3’or 5’(not existent) junction PCR colonies. 

15) When describing the rhBChE functional analysis, why the cells of the transgenic animal were not tested?

16) The Table 2 is not called in the main manuscript.

Some misspellings should be corrected.

Author Response

Here are some concerns that should be addressed by the authors.

1)    Please elucidate how would the protein be recovered from the transgenic animals?

Response: Thank you for this question. As mentioned in figure 3A, both FGF5-sgRNA/Cas9 plasmid and HR arm-CMV-rhBChE template were co-transfected into GFFs. And we generate gene-edited goat that integrated rhBChE at the FGF5 site and we use CMV promoter started rhBChE overexpression. Then rhBChE protein was expressed in the goat.

Since the transgenic animal dead, we didn’t obtain the protein from it. We were planning to get the protein from goat milk according to the methods described in the literature. The production of recombinant proteins by the mammary gland of transgenic animals has been well established in the previous reports (Clark, 1998, J Mammary Gland Biol, 337-350; van Berkel, et al., 2002, Nat Biotechnol, 484-487). A variety of recombinant human proteins, including immunoglobins, growth hormone, and clotting factors have been expressed by the mammary gland and secreted in the milk of transgenic animals. In our manuscript, since the transgenic animal has dead, we planned to get the protein from goat milk samples collected after initiation of induced or natural lactation, which further used to analyze for the presence of the rBChE (Podoly, et al., 2008, Neurodegener Dis, 232-236). 

2)    The main objective of this paper is not clear. Is it to improve HR efficiency? Produce a transgenic animal? Proof of concept regarding the use of CRISPR/Cas system to generate transgenic large animals?

Response: The main objective of this paper is to efficiently generate of gene edited goat. According to previous reports, gene corrections achieved by CRISPR/Cas system often induce an abundance of random insertions and deletions at the target locus because of the presence of DSBs. However, the extremely low efficiency of homologous recombination (HR) and the time-consuming breeding programs were required to obtain genetically modified animals. The progress of genetic modification in large animals is also slow going since zygote injection, combined with inefficient HR, limits the possibility to achieve precise and tailored editing (Zhao, et al., 2019, Natl Sci Rev, 402-420). To solve the problem above, we established an efficient nucleofection system and optimized HA length during HR optimized of GFFs, enabling more efficient precise gene editing in GFFs. Then, we attempted to combine CRISPR/Cas9 and SCNT to generate gene-edited goats that integrated rhBChE at the FGF5 site and promoted rhBChE overexpression through FGF5 knockout.

3)    Please clarify the specie from which BChE was extracted from. I believe it was from a human. Was there an ethic committee involved? If so, please indicate.

Response: Thank you for your question. The gene sequence of BChE was extracted from Homo sapiens. This study was conducted with the approval of the Ethics Committee of China Agricultural University. All experimental methods were carried out in accordance with the approved guidelines.

4)    Be careful when describing units. From lines 103-114, all the measurements were in the milli scale (ml and mg). Is that correct? If so, please elaborate how feasible would be to produce plasmids and DNA template constructs in the “mg” scale. Same thing for line 126, 132-133.

Response: We apologize for our careless writing. Accordingly, we have checked the whole section and corrected the mistake mentioned by the reviewer. See line 107, 110-111, 124-125, 131-132.

5)    During oocyte manipulation (cloning procedure), no protein source was added into the TCM199?

Response: In the oocyte manipulation process, we added 2% FBS in TCM199 medium. To be clear, we listed more details in the medium formulas shown in line 167-173, 186-191.

6)    Please clarify the methodology used for oocyte activation after SCNT. Has the stated protocol (cycloheximide for 5 min followed by cytochalasin B for 5hrs) been used before? If so, please add a reference.

Response: Thank you for pointing out this, we have rewrited the description of this section into “The fused embryos were then incubated in TCM199 with 2% FBS containing 5µM ionomycin (Sigma-Aldrich; C7698) for 5 min and were transferred into TCM199 with 2% FBS containing 2 mM 6-dimethylaminopyridine for 4 hrs. The activated embryos were incubated in TCM199 containing 10% FBS at 38.5°C in a humidified atmosphere of 5% CO2 overnight. These embryos were transplanted into the oviduct of the recipients”. as shown in line 186-191. The protocol has been reported by previous study and we added the reference in the revised manuscript.(Deng, et al., 2013, Theriogenology, 50-57)

7)    Why was melatonin used in different concentrations or sometimes not even used? Do the authors have any results showing the effects of melatonin on embryo survival/transfer efficiency?

Response: We replaced this section into “The fused embryos were then incubated in TCM199 with 2% FBS containing 5 µM ionomycin (Sigma-Aldrich; C7698) for 5 min and were transferred into TCM199 with 2% FBS containing 2 mM 6-dimethylaminopyridine for 4 hrs. The activated embryos were incubated in TCM199 containing 10% FBS at 38.5°C in a humidified atmosphere of 5% CO2 overnight. These embryos were transplanted into the oviduct of the recipients.” As shown in line 186-191.

8)    Did the genetic manipulation of the cell affect embryo development rates? No results were shown regarding in vitro embryo development.

Response: Thank you for this question. Due to the small number of embryos obtained in vitro of our experiment, all of them were used for transgenic operation in this study, so we couldn’t calculate whether genetic manipulation of cells affects the rate of embryo development.

9)    Did the authors had WT SCNT animals born? Were CT embryos transferred to recipient females?

Response: This is a great question. Because we only had eight recipient goats, we did not transplant WT cell lines in order to improve the efficiency of gene editing cell transplantation. However, as an alternate, we chose WT goats of the same breed and age as the control group.

10)   The introduction is very broad and with very overstated phrases, such as lines 73-76. The same happened during the discussion and conclusion. The discussion session does not bring any discussion into the presented results.

Response: Thank you for your suggestion. We have revised the whole manuscript and replaced some broad and very overstated phrases into “In brief, we optimized the efficiency of homologous repair of primary GFFs and improved the productivity of gene-edited goats, demonstrating that the CRISPR/Cas9 system has the potential to become an important and applicable gene-editing tool in large animal breeding.” (Line 73-76).

We also rewrite our discussion section “In addition, we constructed a non-resistant and unmarked knock-in overexpressing an rhBChE-HA donor (Figure 3a) and obtained many positive clonal cell clones by blind screening. Thus, when no marker gene remains in the donor cell, there is no potential risk for the generation of transgenic animals.” Into “In addition, we constructed a non-resistant and unmarked knock-in overexpressing an rhBChE-HA donor and obtained17 positive GFF clones by PCR and sanger sequence screening other than using fluorescence or drug. Due to no marker genes in the donor cells, there is no potential biosafety risk for the generation of transgenic animals.” (445-449).

We also rewrite our conclusion section “Our study provides an avenue to develop the CRISPR/Cas9 system for large animal applications” into “Our study further proves that CRISPR/Cas9 technology has broad applicability in large animal cloning.” (478-479).

11)   Line 269: Figure 1f doesn’t exist.

Response: We have corrected the wrong figure number as shown in line 268.

12)   Figure 2 c and d: odd scale

Response: We have updated figure 2 c and d as shown in line 295.

13)   Please clarify which of the cell clones colony had biallelic modification. Was this the colony used for SCNT?

Response: Thank you for pointing out this. The genomic integration of rhBChE in cell clones were identified by cross-homologous arm PCR using ID-F/R as upstream and downstream primers (amplification product, 3004 bp; Figure 3e, Table S2) in line 316-318. However, with the subsequent expansion culture of clones, we detected that there was no biallelic modification, so we selected 10# clone for embryo transfer. So line 335-337 has been rewrited in our revised manuscript.

14)   Table 1: Please be careful when describing number of 3’or 5’(not existent) junction PCR colonies.

Response: Thank you for your suggestion. We have revised the content in line 347

15)   When describing the rhBChE functional analysis, why the cells of the transgenic animal were not tested?

Response: Due to the traffic control caused by the COVID-19 epidemic in China, transgenic animal cells were not collected in time. Unfortunately, the sheep died at the age of one, with no way to harvest living cells, so we didn’t test the cells of the transgenic animal.

16)   The Table 2 is not called in the main manuscript.

Response: Thank you for careful revision, we have added the table into the main manuscript as shown in line 352.
